# A Beginner’s Guide to Osmoprotection by Biostimulants

**DOI:** 10.3390/plants10020363

**Published:** 2021-02-13

**Authors:** David Jiménez-Arias, Francisco J. García-Machado, Sarai Morales-Sierra, Ana L. García-García, Antonio J. Herrera, Francisco Valdés, Juan C. Luis, Andrés A. Borges

**Affiliations:** 1Chemical Plant Defence Activators Group, Department of Agrobiology, IPNA-CSIC, Avda. Astrofísico Francisco Sánchez 3, 38206 La Laguna, Tenerife, Canary Islands, Spain; fjaviergarma@gmail.com (F.J.G.-M.); algg@ipna.csic.es (A.L.G.-G.); ajherrera@ipna.csic.es (A.J.H.); 2Applied Plant Biology Group (GBVA), Department of Botany, Ecology and Plant Physiology–Faculty of Pharmacy, Universidad de La Laguna, Avda. Astrofísico Francisco Sánchez s/n, 38071 La Laguna, Tenerife, Canary Islands, Spain; saraimoralessierra@hotmail.com (S.M.-S.); fvaldes@ull.edu.es (F.V.); jcluis@ull.edu.es (J.C.L.)

**Keywords:** osmotic stress, drought, biostimulants, osmoprotectants, salinity

## Abstract

Water is indispensable for the life of any organism on Earth. Consequently, osmotic stress due to salinity and drought is the greatest threat to crop productivity. Ongoing climate change includes rising temperatures and less precipitation over large areas of the planet. This is leading to increased vulnerability to the drought conditions that habitually threaten food security in many countries. Such a scenario poses a daunting challenge for scientists: the search for innovative solutions to save water and cultivate under water deficit. A search for formulations including biostimulants capable of improving tolerance to this stress is a promising specific approach. This review updates the most recent state of the art in the field.

## 1. Introduction

In the next 30 years, world population is expected to increase by 2 billion, reaching 9.7 billion in 2050 [1]. This rate of increase is expected to double during the early 21st century, recently being 25–70% [2]. That scenario has focused researchers and international organizations on the need to ensure adequate food production, to feed the growing population. A plethora of recent studies have projected situations for the coming years, especially in economically less developed regions such as the Middle East [3] and Latin America [4,5], but also in “rich” countries [6]. Climate change is probably the most important factor endangering agricultural production, especially in Mediterranean and African regions. There, drought periods will almost certainly intensify, yield in more arid zones being intrinsically linked to water availability [7]. Indeed a decrease in productivity due to environmental stress can be forecasted for the upcoming years [8]. Discussion or controversy has arisen in response to this widespread perspective of a need to increased food production to ensure the world’s food supply [2]. Its opponents in the debate claim this is not necessary because current global production is sufficient to meet human nutritional needs [9]. Nevertheless, extreme environmental conditions certainly endanger future food security.

Drought and salinity together reduce global crop production by as much as 50%, and their impacts are indeed intensified by climate change [10]. These two factors exert osmotic stress on plants through a drop in soil water potential, due to restricted water availability or solute concentration, respectively (Figure 1).

They cause similar responses, causing water deficit, nutrient imbalance and oxidative stress. The plant responds quickly to this assault by closing its stomata to prevent water loss, which limits gas exchange and consequently photosynthesis and growth [11]. To counteract this, plants need to adjust osmotic balance, this being the main physiological adaptation mechanism they use to cope with osmotic stress. They do this by accumulating different kinds of solutes, such as amino acids (especially proline), glycine betaine, and sugars [12]. These osmolytes protect cell structure and function, facilitating water uptake and retention [13]. Since they induce tolerance against osmotic stress [14,15,16], they are called osmoprotectants [17]. Therefore, the study of these osmolytes and other substances able to induce tolerance [18] is a promising research field. It operates within the search for environmentally friendly treatments to increase food production under stress. Through the upcoming “Farm to Fork” strategy, the European commissioners recommend that EU members work using ecological and sustainable methodologies, reducing fertilizers and pesticides by 30% and 50%, respectively, by 2030 [19]. In this regard, biostimulants are defined by EU regulation [20] as: “*A product that stimulates plant nutrition processes independently of the product’s nutrient content, with the sole aim of improving one or more of the following characteristics of the plant or the plant rhizosphere: (a) nutrient use efficiency; (b) tolerance to abiotic stress; (c) quality traits; or (d) availability of confined nutrients in the soil or rhizosphere*”. Biostimulants include various kinds of substances, categorized in different ways over the years [21]. Du Jardin [22] classified them into seven different categories: Humic/Fulvic acids, Seaweed/Botanical extracts, Protein hydrolysates, Biopolymers, Beneficial minerals, Beneficial bacteria, and Beneficial fungi. Later on, this system was revised by Bulgari et al. [23], who added another category including extracts from industrial or food wastes and grouped nanomaterials and nanoparticles into the category of biopolymers. Recently, we focused attention on a group of pure organic products cited in the literature as having the proven ability to improve plant tolerance against abiotic stress [18], regardless of the previously established classifications of biostimulants.

The global market in biostimulants is expected to reach $4.14 billion by 2025 [24]. In fact, many companies have already developed formulations to improve productivity under stress. These usually consist of extracts or hydrolysates from various sources [25,26], mixed with beneficial compounds acting alone as osmoprotectants [18]. This illustrates how important it is to consider the synergic effects between several molecules acting together. Indeed, it should be the best approach to combat osmotic stress, rather than the idea of finding a “magic bullet” capable of mitigating the insults alone. The aim of this review was to examine the literature and present a wide range of biostimulant sources (Figure 2) with a demonstrated capacity to increase tolerance to osmotic stress. We group them into three main categories: (1) naturally accumulated plant osmoprotectants; (2) natural and non-natural plant protectant compounds; and (3) hydrolysed biological extracts and microorganisms. This particular classification may help researchers that are beginning to delve into the field of biostimulants.

## 2. Naturally Accumulated Plant Osmoprotectants

A number of bacteria, seaweeds and plants are able to accumulate several types of compounds called compatible solutes or osmoprotectants to face osmotic stress, for instance amino acids (proline, glutamate, etc.), carbohydrates (trehalose), sugar alcohols (inositol, mannitol), quarternary ammonium compounds (glycine betaine) [27] and tertiary sulphonium compounds (e.g., dimethylsulphoniopropionate) [28].

### 2.1. Amino Acids: Their Involvement in Osmotic Adjustment

Amino acids are probably the group of compounds most used in biostimulation, offering a solid correlation with improvements in coping with osmotic stress [29,30]. Commercially, the normal way to achieve this is to use protein hydrolysates from a diverse range of sources. For an interesting review focused on commercial biostimulant formulation, see Madende and Hayes [24]. We wish to point out several amino acids that interact within osmotic adjustment in plants and can clearly induce tolerance.

Proline is probably one of the most studied amino acids acting against stress. It is accumulated in eubacteria, protozoa and plants, and is reported to aid in facing drought and salt stress by balancing the adverse osmotic potential that prevents water uptake [31]. Proline biosynthesis thus assists plants in the acclimation process. Two such pathways are described in plants: ornithine or glutamate. The glutamate pathway seems to be controlled by osmotic stress [32]. Specifically, it begins with pyrroline-5-carboxylate synthetase (P5CS) that uses ATP and NADPH to reduce initial glutamate into glutamate-semialdehyde, which converts spontaneously to pyrroline-5-carboxylate [33]. P5C is reduced to proline by the action of P5C reductase (P5CR), using NADPH and H^+^ [33]. In most plant species, P5CS is encoded by two genes, *P5CS1* and *P5CS2*, while *P5CR* is encoded by only one [33]. Proline metabolism is particularly interesting owing to the intracellular localization of P5CS in chloroplast and cytoplasm. Proline biosynthesis requires using high amounts of NADPH and ATP. In chloroplasts, it is thought to contribute to maintaining a low NADPH:NADP^+^ ratio, thus sustaining electron flow in the photosynthetic chain, stabilizing redox balance, and finally reducing photoinhibition and consequent damage to the photosynthetic apparatus [34]. The osmotic stress degradation pathway is down-regulated, ensuring free proline accumulation. After stress, proline is catabolized in mitochondria, supporting oxidative respiration with energy to resume growth after stress. Indeed, complete oxidation of proline would yield 30 ATP molecules (Figure 3). Therefore, proline reserves are valuable not only in osmotic adjustment during acclimation, but also to facilitate recuperation after stress [35].

Osmotic stress induces genes involved in proline biosynthesis, which leads to proline accumulation. This was demonstrated by Székely et al. [36], who knocked out p5cs1 in *Arabidopsis* plants to produce a salt-sensitive mutant. Exogenous application of proline can improve tolerance to salt stress through regulation of the endogenous proline metabolism. For instance, foliar application of proline to maize resulted in decreased P5CS activity and an increase in PDH under salt stress [32]. In accordance with the attributes to be expected from its metabolism, proline treatment is capable of alleviating lowered photosynthetic activity and enhancing water relations under salt stress in *Olea europaea* L. cv Chemlali [37]. Moreover, it can stabilize mitochondrial electron transport complex II, membranes, proteins and enzymes such as RUBISCO [38]. In *Sorghum bicolor* [39], it was also shown how this proline treatment increases growth under stress conditions. Finally, as a further example of ongoing research, Abdelaal et al. [40] illustrated how proline can increase production under salt stress. Indeed, altogether the weight of evidence demonstrates the osmoprotectant effect of exogenous proline treatment.

Glutamate is a central molecule in amino acid metabolism in higher plants. In this regard, the α-amino group of glutamate is directly involved in assimilation and dissimilation of ammonia and is transferred to all other amino acids, since it is the precursor of γ-aminobutyric acid (GABA), arginine, and proline. It should also be noted that glutamate is the precursor for chlorophyll synthesis in the developing leaves [41]. Plants can synthesize glutamate by various pathways, principally via the glutamine synthetase/glutamine-α-oxoglutarate transaminase cycle in the chloroplast. Otherwise in non-photosynthetic tissues, they can proceed via glutamate dehydrogenase in the mitochondria or cytoplasm. Finally, plants can also produce glutamate by the alternative pathways pro/pyrroline 5-carboxylate cycle and transamination [42]. Glutamine synthetase/glutamine-α-oxoglutarate transaminase is the key cycle for assimilation of ammonium in plant cells. However, in tobacco plants under salt stress, its activity did not change significantly, but glutamate dehydrogenase activity increased. This is a stage in the glutamate synthesis pathway under stress, where it acts as a proline precursor [43]. These results were partially supported by Wang et al. [44]; since at low salinity the authors found that glutamine synthetase was the preferred active enzyme increasing photorespiration, while at high salt concentration it was glutamate dehydrogenase.

Glutamate treatment is rapidly converted to other amino acids, such as glutamine, GABA or proline in rice roots [45]. Under salinity stress, treatment with glutamate acts as a precursor of proline to cope with salinity stress [46]. In fact, foliar treatment of *Brassica napus* L. under drought stress with glutamate can increase *P5CS* expression [47], leading to proline accumulation and better osmotic adjustment.

γ-aminobutyric acid (GABA) is a widely distributed non-proteinogenic amino acid and a significant component of the free amino acid pool in bacteria, fungi, plants and animals. Under stress, GABA production increases, quickly reaching higher concentrations than other amino acids involved in protein synthesis [48]. Its biosynthesis begins from glutamate by action of glutamate decarboxylase, and accumulation leads to H^+^ or Ca^2+^/calmodulin, which activates the enzyme to yield GABA (Figure 3). The amino acid can be exported to the mitochondria and, after several enzymatic steps, transformed to succinate, the main energy source for the tricarboxylic acid cycle [49,50]. This route is known as the GABA shunt and links primary and secondary carbon and nitrogen metabolisms [42,51]. Interestingly, Shelp et al. [52] found that GABA transaminase activity (the first enzyme involved in the shunt) was inhibited under stress conditions. This suggests that GABA accumulation would be useful for the provision of anaplerotic succinate for the Krebs cycle, while awaiting the end of the stress. These data are supported by trials with GABA-deficient mutants her1 and gaba-t/pop2-1, in which no GABA-derived succinate was found in roots after salt stress ended [53,54].

The involvement of GABA in osmotic regulation was shown by using the mutant line gad1/2 in Arabidopsis. In response to water deficit, its leaves had lower relative water content (RWC), higher abaxial and adaxial stomatal conductance, and wider stomatal opening, compared to the wild type [55]. In addition, GABA treatments are able to increase tolerance of osmotic stress in rice [56], improving ionic and redox balance to reach better osmotic adjustment by accumulation of organic osmolytes such as proline, sugar and starch.

Arginine is an interesting amino acid in plant metabolism, since it has the highest nitrogen/carbon ratio of the 21 proteinogenic amino acids, and is apparently a precursor of NO and also a well-documented precursor of polyamines. Synthesis in chloroplasts via ornithine is apparently the only operational pathway to provide arginine in plants, although in fact arginine biosynthesis is poorly studied in plants [57]. One of the products of arginine hydrolysis is ornithine, by means of ornithine δ-aminotransferase, which is hypothesized to play an important role in osmotic adjustment under stress. However, ornithine δ-aminotransferase activity has been correlated with proline accumulation in salt-stressed plants [58]. Overexpression of this enzyme in rice resulted in higher proline levels and activated antioxidant defence, thus enhancing stress tolerance [59].

In particular, arginine treatment improves antioxidant defences in plants; indeed, Nasibi et al. [60] researched into a foliar application that enhanced the detoxification response, lowering H_2_O_2_ content under water stress. External treatment can also increase proline accumulation in mung bean under salinity stress, protecting its growth [61].

### 2.2. Carbohydrate Sugars 

Carbohydrates are crucial molecules for living organisms. Chemically they are aldehydes and ketones with several hydroxyl groups and varying degrees of polymerization (monosaccharides, disaccharides, oligosaccharides and polysaccharides), containing atoms of carbon, hydrogen, and oxygen [62]. They exert structural, signalling, carrying and storage functions. The part they play in the plant’s response to stress has been associated with their high potential for polymerization. Due to their solubility, some of them: sugars such as hexose and fructans (and also sugar-alcohols like mannitol and sorbitol—see below) can act as compatible solutes, contributing to osmotic balance and membrane and protein stabilization under osmotic stress conditions [62,63]. These substances clearly act as osmoprotectants [64], since a number of studies have demonstrated that accumulation of reduced forms of sugars has an osmoprotective function against drought and salt stress [65].

Starches are also emerging as key substances in mediating plant responses to abiotic stresses, such as water deficit and high salinity. They consist of two types of molecules: linear and helical amylose and the branched amylopectin. Depending on the plant, starch generally contains 20 to 25% amylose and 75 to 80% amylopectin by weight. Under challenging environmental conditions, plants generally remobilize starch to provide energy and carbon at times when photosynthesis becomes limited. It is therefore used for many plant species as an acclimation strategy in harsh environments [66]. Fructans are considered short-term storage carbohydrates and have been associated with stress tolerance mechanisms for many years [67]. This is especially the case with freezing tolerance, since desiccation is a key component in some research focused on studying the possibility to alleviate drought stress using fructans. On this topic, Su et al. [68] showed how the 1-fructosyl-transferase gene from a fructan-accumulating plant increases tolerance against osmotic stress induced by PEG in tobacco plants. These fructans may indirectly contribute to osmotic adjustment by releasing hexose sugars [67]. Raffinose family oligosaccharides are characterized as compatible solutes involved in stress tolerance, and are accumulated under drought stress [69]. However, despite their osmoprotective action, we have not found reports of these compounds used as biostimulants.

Soluble sugars such as sucrose, glucose and fructose are important not only as nutrients but play other roles in metabolism, growth and stress responses [70]. Their accumulation as osmolytes serves to help plants face the negative effects of osmotic stress [71]. As an example, treatment with glucose enhances salinity tolerance in wheat seedlings by preventing water loss through proline accumulation and maintaining ionic balance [72]. It was earlier shown that soluble sugar accumulation enhances proline content and thus helps the plant to counteract the osmotic insult of salt stress [73].

Trehalose is a promising carbohydrate for use as a biostimulant. It is a non-reducing disaccharide consisting of two glucose units (α-D-glucopyranosyl-1,1-α-D-glucopyranoside) and is widely spread in a variety of organisms: bacteria, yeast, fungi, lower and higher plants, as well as insects and other invertebrates [74]. Its biosynthesis in plants occurs in two steps: trehalose-6-phosphate synthase generates trehalose-6-phosphate from **uridine diphosphate** UDP-glucose and glucose-6-phosphate, followed by dephosphorylation to trehalose by trehalose-6-phosphate phosphatase [75]. Some work using genetic engineering provides evidence that an enhanced metabolism can positively regulate osmotic stress tolerance. As an example, Ge et al. [76] showed how over-expression of OsTPP1 caused higher tolerance to salt stress in rice. See Fernandez et al. [77] for an extensive review of the action of trehalose in stressed plants. Trehalose can be used as a treatment to enhance tolerance against salt stress in strawberry [78], protecting plants from oxidative damage caused by salt and conserving the photochemical function. Moreover, trehalose treatment is capable of enhancing osmoregulation in wheat.

### 2.3. Sugar Alcohols

Also known as polyols, sugar alcohols can be structurally cyclic like myo-inositol or have a linear structure such as mannitol or sorbitol [79]. Usually water soluble, polyols are derived from the reduction of aldoses or their phosphate esters [65]. In particular, sorbitol, a sugar alcohol with six carbons, is widely distributed in plants. It is a major photosynthetic product in apple trees, where foliar treatments suppress its synthesis and alter the stress expression profile. This suggests that sorbitol therefore plays an important part in responses to abiotic and biotic stresses in apple trees [80]. Other sorbitol treatments to elicit stress reactions remain out of the scope of this review.

Inositols are synthesized from D-glucose, involving three enzymatic steps. Hexokinase converts glucose into glucose-6P and then Ins(3)P1 synthase produces Ins(3)P1, which is the first step in myo-inositol (MI) biosynthesis. Finally, phosphate loss due the action of MI monophosphates releases free MI. This constitutes the pathway for MI biosynthesis in cyanobacteria, algae, fungi, plants, and animals and is central to their cellular metabolism [81]. Plants accumulate many kinds of inositol during abiotic stress periods caused by drought and high salinity stresses [82]. In this context, manipulation of inositol metabolic pathways can increase salt tolerance in rice [83]. Myo-inositol treatment helps to maintain cell turgor, enhancing water status in Capsicum anuum [84]. 

Mannitol has a wide presence in plants and fungi, with importance in the tolerance response against osmotic stress [85]. It is biosynthesized in the plant cell cytoplasm from fructose-6P, which is transformed by phosphomannose isomerase into mannose 6-phosphate, then mannose-6-phosphate reductase transforms it into mannitol 1-phosphate. Finally, mannitol-1-phosphate phosphatase removes the phosphate to yield mannitol [85]. Abebe et al. [86] introduced the ectopimannitol-1-phosphate dehydrogenase from *Escherichia coli*, which enabled mannitol accumulation in wheat calluses (a plant that does not accumulate mannitol). It increased tolerance to osmotic stress and oxidative impacts by preventing water loss. This effect can be elicited using an external treatment of mannitol in wheat [87], and in maize where it can enhance proline metabolism under drought stress, improving water relations [88].

### 2.4. Quaternary Ammonium Compounds

Quaternary ammonium compounds accumulated in plants are glycine betaine, β-alanine betaine, proline betaine, choline-O-sulphate, hydroxyproline betaine and pipecolate betaine [28]. These organic compounds are known to also have osmoprotective effects in plant cells [89]; they are thus an interesting family to consider for biostimulant formulations.

Glycine betaine is accumulated by numerous organisms, such as bacteria, cyanobacteria, algae, fungi and animals. It is probably the most common quaternary product that plants accumulate to cope with osmotic stress [28]. Glycine betaine is synthesized preferentially from choline, which is converted to betaine aldehyde by choline monooxygenase. Then, it is converted by the action of betaine aldehyde dehydrogenase into glycine in all glycine betaine-accumulating plant species [28]. Some plants, such as aubergine/eggplant, potato, *Arabidopsis*, tomato and many rice cultivars, cannot accumulate detectable amounts of glycine betaine [90]. Therefore, genes associated with its biosynthesis have been introduced/overexpressed in these non-accumulating plants, e.g., in tomato [91] or potato [92], demonstrating the fundamental role of glycine betaine in osmoprotection against stress. Overaccumulation of this osmolyte provides tolerance in wheat against drought, due to upregulation of the betaine aldehyde dehydrogenase gene [93]. Glycine betaine is able to induce tolerance to abiotic stress after treatment, for example, in maize, modulating the ABA response against drought and salt, and preserving yield [94]. A wide number of species show better performance against drought and salt stress after treatment [16].

L-proline-betaine (also called stachydrine) is accumulated in non-halophytic *Medicago* species [95], although it is less frequently present in halophytes than glycine betaine. Interestingly, some researchers point out that proline betaine accumulation is an evolutionary response to salinization, deriving from several proline methylation steps [95]. Treatments applied in *Bacillus subtilis* cultures increase tolerance against osmotic stress [96] and have more effective osmoprotectant effects than proline [97]. However, as far as we know, proline-betaine has not been assayed as a plant treatment to alleviate stress, despite being a promising compound for study as a biostimulant. Finally, we must highlight that alanine-betaine acts as a compatible osmolyte in halophytic Plumbaginaceae species; this constitutes another interesting evolutionary adaptation against combined osmotic and sulphate stresses, for example. Its biosynthesis does not require oxygen (in contrast to glycine betaine), and its use may be advantageous in sulphate-rich salt marsh environments [98].

### 2.5. Tertiary Sulphonium Compounds

These substances are an interesting family of compatible osmolytes, scarcely studied in comparison to other groups and of course even less as biostimulants. For instance, the known anti-stress compound dimethylsulphoniopropionate (DMSP) is synthesized in many algae but only a few plants, notably genus *Spartina* and in sugarcane [99]. It is also accumulated in some cyanobacteria and bacteria [100]. The precursor of DMSP is methionine and its involvement in osmoregulation is based on structural similarity with quaternary ammonium compounds. There are some studies on the osmoregulatory function of DMSP in algae, and the data presented in *Spartina* species provide evidence to consider it an osmoprotectant [99]. Another compound from this family is S-methylmethionine (vitamin U), produced by all angiosperms. Treating plants with it to enhance cold tolerance has been studied [101,102], but to date its application against salt or drought stress has not been. 

## 3. Natural and Non-Natural Plant Protectant Compounds

Several treatments are capable of increasing plant tolerance to such stresses, such as those with polyamines [103], silicon [10], menadione sodium bisulphite [104], and melatonin [105,106]. These compounds are promising options to aid in further understanding the response mechanism under osmotic stress or also biostimulation processes.

### 3.1. Polyamines

These are nitrogen-containing aliphatic compounds with low molecular weights. At the physiological level they are positively charged, regulating pH among cellular components [107]. These compounds bind electrostatically with great affinity to macromolecules, such as proteins, DNA, and RNA [108]. As important examples, putrescine, spermine, spermidine, and other phytohormones that contain aliphatic amines are found in bacteria and animals, besides plants [109]. Polyamines (PAs) play a key part in essential biochemical/physiological processes: development, cell proliferation, signal transduction and senescence. They are important in gene expression, and respond to stresses in eukaryotic and prokaryotic cells. Several papers confirm that plants accumulate an enormous quantity of PAs to face a range of both abiotic and biotic stresses [109,110].

Polyamine syntheses begin with the formation of putrescine, for instance, by direct decarboxylation of ornithine through the activity of ornithine descarboxylase. However, putrescine can also be synthesized from arginine by decarboxylation into agmatine by action of arginine decarboxylase, followed by agmatine iminohydrolase and N-carbamoyl putrescine aminohydroxylase to produce putrescine [111]. Spermidine is formed from putrescine by the action of the spermidine synthase, and finally spermine is catalysed through the intervention of spermine synthase [111]. 

External polyamine treatments can increase tolerance against osmotic stress:Putrescine is capable of protecting photosynthetic machinery in cucumber under salt stress [112]. It can also improve gas exchange parameters [113].Spermidine improves drought tolerance in maize [114] by strengthening antioxidant defences. Under salt stress it enhances reactive oxygen species (ROS) scavenging to promote tolerance [115].Spermine increases tolerance to salt stress in tomato by enhancing the chloroplast antioxidant system [116]. Drought effects can be mitigated by spermine treatment in maize [117], alleviating photosynthesis inhibition.Farooq et al. [118] researched into treatments applying the three polyamines to rice, to ameliorate drought stress. Their study showed how the best results in controlling water loss were obtained by foliar application of spermine. Nevertheless, putrescine and spermidine had superior results in scavenging ROS radical, with a better enzymatic response.Thermospermine is synthesized by thermo-spermine synthase and is less well known than the other three polyamines. It has not been used as a treatment against stress, but its potential is worthy of mention because a thermospermine-deficient mutant is hypersensitive to salt in *Arabidopsis thaliana* [119].

### 3.2. Silicon

The physiological roles of silicon in animals have been known for more than a century. Nevertheless, its specific benefits to plants, particularly under stress, have only recently been under intensive study. This is mainly due to it being labelled a “non-essential" element by plant nutritionists [120]. Certainly, Si is not deemed “essential" to vascular plants, since they can carry on their life cycles in its absence. Some plants are silicon accumulators, such as rice and sugarcane [121]. Interestingly, other plants defined as non-accumulators have beneficial effects after silicon treatment. As an example, tomato is not a silicon accumulator; moreover, it is considered an excluder, but it is clearly established that this metalloid can increase tolerance against salt stress [122]. 

Silicon can be applied in several forms: K_2_SiO_3_ has been applied in wheat to alleviate drought stress [123], in common bean to enhance salt stress tolerance [124], and can increase yield under drought stress [125].Na_2_SiO_3_ is able to improve salt stress tolerance in barley [126] and in maize growing under drought stress [127]. However, as far as we know, increased production under stress was not reported using this silicon compound.SiO_2_ is applied using micro- or nanoparticles to alleviate the deleterious effects of drought in rice [128] and salinity in potato. Indeed, it is possible to increase yield under drought stress using these nanoparticles [129].H_4_SiO_4_ can induce drought tolerance, increasing yield and enhancing fruit quality in watermelon [130].

Silicon aids the plant to cope with osmotic stress in various ways. Two processes contribute to stress tolerance: (i) mechanical and physical protection due to SiO_2_ deposits known as phytoliths; (ii) biochemical responses that trigger metabolic changes [131]. Silicon can influence water relations in plants submitted to drought, reducing water lost through cuticular transpiration and stomatal conductance [132]. Interestingly, transpiration rates decrease with increasing Si content in shoots [133]. Directly counteracting salinity stress, it can raise Na and Cl uptake [132,134].

### 3.3. Vitamin K_3_ (Menadione Sodium Bisulphite)

This vitamin is erroneously thought to have only a synthetic origin, whereas it can be isolated from fungi, cryptogams and phanerogams, although its functions are still unclear [135]. Our research group has studied in depth the anti-stress properties of a water-soluble derivative called menadione sodium bisulphite (MSB). Besides its capacity to induce resistance against many plant pathogens and pests [136,137], this molecule increases tolerance to seed treatment against salt stress in *Arabidopsis* [138], by root treatment [139], or foliar application [140]. Treatment under salt stress triggers a slight oxidative burst that elicits plant defences that lead to a faster relative growth rate, with enhanced water status and gas exchange parameters [139]. Interestingly, treatment demethylated the promotor of the *P5CS* gene in *Arabidopsis*, causing an earlier response in proline metabolism and better osmotic adjustment [141]. Furthermore, MSB also enhanced scavenger responses and prevented toxic Na^+^ levels by the expression of regulating proteins, thus improving ionic homeostasis under salt stress [139]. Beyond the clearly demonstrated greater tolerance to salt stress, foliar treatment with MSB also slightly improved drought stress response during the first steps of stress exposure in broccoli [142]. Additionally, MSB has been shown to induce tolerance against heavy metals such as cadmium [143] and chromium [144]. 

### 3.4. Melatonin

A compound almost universally present in animals, melatonin is also in plants, where it is distributed in various organs, such as leaves, stems, roots, fruits and seeds [145]. In a wide range of plant species, melatonin biosynthesis begins with tryptophan being transformed to tryptamine by tryptophan decarboxylase, then tryptamine 5-hydroxylase catalyses the conversion of tryptamine to serotonin. Serotonin is converted to N-acetyl-serotonin by serotonin/arylalkylamine N-acetyltransferase, and finally into melatonin by N-acetyl-serotonin /hydroxyindole O-methyltransferase [146]. Melatonin is involved in numerous plant metabolic processes (for an extensive review, see Back [147]) and has a prolific bibliography, particularly due its involvement in enhanced plant tolerance to abiotic stress.

Under unfavourable environmental conditions, plants accumulate melatonin [148]. In this regard, overexpression of the genes involved in melatonin biosynthesis enhances drought tolerance [149], confirming its implication in plant defences against osmotic stress. Exogenous treatment with melatonin also induces resistance against salt stress in maize, ameliorating oxidative stress and adjusting ion balance [150]. Similarly, the effect of exogenous melatonin on the antioxidant system can increase drought tolerance in kiwi-fruit seedlings [151]. In addition, the treatment impedes water loss in stressed plants. Interestingly, 2-hydroxymelatonin is more common in plants, the average ratio being approximately 368:1, but the compound predominantly used as a biostimulant is melatonin. Despite this, some authors have demonstrated the former’s less studied capacity to induce tolerance to drought stress [152].

## 4. Extracts from Natural Sources and Microorganisms

In biostimulant formulation, manufacturers normally use complex mixtures from a range of sources. In addition to the compounds described above, in this review, we must include seaweed extracts, microorganism-based biostimulants and humic and fulvic substances.

### 4.1. Seaweed Extracts

In recent decades, after a notable boom in 1947 in the United Kingdom [153], the use of macroalgal extracts as a source of biostimulants for agriculture has again been on the rise commercially, in the form of plant-growth-promoting factors to enhance salinity, drought and heat tolerance. These preparations act on several biochemical pathways to improve stress resistance. Among these, they induce ROS-scavenging enzymes, membrane stability and an increase in osmoprotectant compounds such as Pro and glycine-betaine [153].

Red, green, and brown seaweeds amount to 10% of the sea’s biomass productivity. Currently, a wide number of companies process and market their extracts for agriculture as biostimulants, which leads to reduced consumption of mineral amendments with their heavy carbon footprint [154]. The most commonly used extraction methods are mechanical disruption, alkalis or acids, and pulverization [153], but other novel techniques could lead to further final biostimulants. It is therefore necessary to evaluate not only the source but also the methodology, to provide a consistent biostimulant effect on crops [155]. Even though the growth-promoting effects of such extracts are reported for a range of crops, the mechanisms underpinning their influence are still not known. The complex chemistry of macroalgae makes it hard to establish exactly which are the active components [156]. In fact, algae have a plethora of osmoprotectant compounds that are able to increase tolerance against stress, such as: ions like Ca^2+^, amino acids, betaines, carbohydrates, flavonoids, phenolic acids, phytohormones, polyols, and tertiary sulphonium [157]. We need to point out that seaweed extracts are an excellent source of phytohormones [158] and that water stress reduces cytokine levels and increases ABA biosynthesis in plant tissues. Cytokinins are in fact key phytohormones that not only regulate plant growth and development but also mediate plant tolerance to drought stress [159]. Seaweed extracts thus act positively on chlorophyll synthesis, keeping cytokinin concentrations high and hindering ABA synthesis.

The algal extracts most used as biostimulants are the following:*Ascophyllum nodosum* has a demonstrated effect as biostimulant (see Shukla et al. [160], for a complete review), being able to enhance tolerance against drought stress, for example, in tomato [161] and avocado plants exposed to salt stress [162]. Under salinity, extracts containing this seaweed can improve water relations to cause better growth and fruit quality in tomato; although yield improvements are not yet reported [163]. Treatments with *A. nodosum* extracts provide drought stress tolerance to treated plants by reducing stomatal conductance and cellular electrolyte leakage in water-stressed plants over time, and maintaining a high leaf turgor and stem angle [164].*Ecklonia maxima* is another macroalga used in biostimulant formulation; research was largely focused on phytohormone-like activities [165], but its effects on stress are less studied. In the literature, however, formulates based on this source are capable of inducing salt tolerance, enhancing gas-exchange parameters and increasing yield in zucchini squash (courgette) [166].

Other species are claimed by Sharma et al. [154], as source biostimulants. Mattner et al. [167] demonstrated how the use of a mixture with *Durviella* and *A. nodosum* can increase yield in strawberry without imposed stress. An extract from *Macrocystis pyrifera* enhanced growth in *Lactuca sativa* seedlings [168] and other interesting species without specific biostimulant uses reported in the literature which include: *Durvillea antarctica*, *Fucus serratus*, *Himanthalia elongata*, *Laminaria digitata*, *Laminaria hyperborea*, and *Sargassum* spp. 

In recent years, biostimulants based on microscopic algae have taken on special importance, with many species acting as biostimulants: *Chlorella vulgaris*, *Chlorella ellipsoida*, *Chlorella infusionum*, *Acutodesmus dimorphus*, *Scenedesmus platensis*, *Scenedesmus quadricauda*, *Dunaliella salina*, *Spirulina maxima* and *Calothrix elenkinii* (see Ronga et al. [169] for an extensive review). Biostimulants using microalgae present several challenges to solve, due precisely to their novelty. Clearly, the great variability of microalgal strains yet unexploited by the biostimulant industry, and lack of knowledge regarding their biomolecular mechanisms, may still cause rejection by professionals and hamper their regular utilization in agricultural practices [170]. 

### 4.2. Microorganisms

It is widely accepted that plants can maintain stabilizing relationships with bacteria that help them survive numerous stressful conditions [171]. In this context, numerous researchers have attempted to isolate microorganisms from varied ecosystems with environmental constraints, such as saline, alkaline, acidic, and arid soils [172]. Microorganisms that grow in adverse environments have had to develop numerous strategies to survive, for example, changes in cell wall composition and high concentrations of soluble solutes. Consequently, these microorganisms are excellent sources of biostimulants [172] and such microbe-induced tolerance is being used against biotic and abiotic stress and enhanced soil fertility [173]. Some genera used as biostimulants are as follows:*Rhizobium* has been described as helping plants to acclimate to abiotic stress [174]. Symbiosis with *Rhizobium* is reported to influence salt and drought responses in *Medicago trunculata* [175].*Trichoderma* is described by Zaidi et al. [176] as a potential source for abiotic stress biocontrol; indeed, exogenous treatment can induce tolerance against salt in wheat [177] and cucumber [178]. It has a capacity to induce tolerance against drought in maize [179] and rice [180].*Bradyrhizobium* can alleviate salt stress, promoting symbiosis in soybean [181]. In peanut, it helps to alleviate the negative effects of water restriction [182].*Azotobacter* isolated from semi-arid regions is capable of alleviating drought stress after inoculation in maize [183]. It is described by Viscardi et al. [184] as a possible way to increase tolerance against abiotic stress.*Azospirillum* can improve salt tolerance in chickpea, increasing the biosynthesis of compatible osmolytes and enhancing the antioxidant machinery [185]. Utilization of this genus was reviewed by Vacheron et al. [186].*Pseudomonas* induces drought tolerance in mung bean [187] and salt tolerance in cotton plants [188].*Bacillus* improves drought stress tolerance in maize, promoting better oxidative and water balance [189]. It is noteworthy that Li et al. [190] studied the transcriptome profile of salt tolerance conferred by a *Bacillus* microorganism.

Treatment with microorganisms may be a useful way to cope with new upcoming situations imposed by climate change [191].

### 4.3. Humic and Fulvic Acid Extracts

These make up more than 60% of soil organic matter and are the major component of organic fertilizers, produced by the biodegradation of organic matter, resulting in a mixture of acids containing phenolate and carboxyl groups. Fulvic acids are humic acids with a higher oxygen content and lower molecular weight [172]. One hypothesis about the mechanism of action of supplementation with these biostimulants in soils with poor organic carbon is that it helps improve microorganism stabilization and chemical characterization of the soil [192]. These authors also found that it is not economically feasible to apply them to arid soils. In addition, published data show that humic and fulvic treatments can modify the plants’ primary and secondary metabolism against abiotic stress, enhancing water uptake and antioxidant behaviour under stress [193]. The application of both types of acids is widely reported in the literature as an external treatment to increase tolerance against osmotic stress (see Ali et al. [128,194] for an extensive review):Humic acids are able to enhance maize plants’ salt response [195], increasing proline accumulation and strengthening the enzymatic antioxidant system. In Lima bean [196] it can ameliorate negative effects exerted by drought, by increasing photosynthetic activity and accumulating sugars and proline in leaves and thus a higher relative water content in these organs.Fulvic acid treatment enhances water relations in citrus through higher proline accumulation in leaves and protection from chlorophyll degradation by salt stress [197] In rapeseed it can protect the photosynthetic machinery and aid the membrane to resist peroxidation [198].

Use of humic substances as a biostimulant for plant growth is a promising eco-friendly approach, in accordance with the concept of circular economy. It focuses on a progressive conversion to resources whose consumption can itself alleviate anthropic impacts and pressure, and the impending consequences of climate change [194]. Finally, there is interesting research where the authors try to isolate humic substance structures and associate them with their activity in plants [199,200], laying the basis for a better understanding of this structurally complex type of biostimulants.

## 5. Biostimulants’ Field Applications

The European Biostimulant Industry Council also reported that more than 6.2 million hectares in the EU have already been treated with biostimulants. Developments in this field are definitely focused on looking for new compounds or mixes that can be applied to increase and economize crop production by reducing costs. Yakhin et al. [21] provide interesting information about one of the most famous biostimulants on the market. 

Biostimulants can be administered in the field by foliar or fert-irrigation, for example, amino acids are normally applied as protein hydrolysates. This kind of product may proceed from animal or plant sources. They are quickly gaining popularity in the industry and with farmers, because they contain a large number and quantity of bioactive compounds and have proven efficacy in enhancing crop performance, even under stress conditions [26]. Biostimulants based on amino acids are suitable for foliar application, increasing yield and grain quality after two treatments with these commercial formulations [201]. Foliar treatments are also able to increase yield under saline growth conditions [202] or growth under water deficit conditions [203,204]. Amino acid formulations can also be applied by fertirrigation to enhance fruit quality [205], improving yield under water restriction [206]. In an interesting review, Moreno-Hernandez et al. [207] summarize commercial biostimulants prepared from protein hydrolysates in a table showing their application method and crop responses. Basie et al. [208] review how biostimulants from different sources have been applied to fruit trees, grapevines and berry crops against abiotic stress, including drought and salinity. In an interesting field experiment, Kocira et al. [209] used foliar treatments of seaweed- or amino-acid-based biostimulants. Both increased yield, but the profits using seaweed extracts are higher in comparison to amino acids: EUR 752.57·ha^−1^ to EUR 119.67·ha^−1^, respectively (Figure 4). 

## 6. The Future of Osmoprotection

### 6.1. What Chemical Modifications Can Be Learned from Nature and Applied to Single Molecules?

Plants are incredibly resilient organisms that have to live their whole life cycle in the place that a seed or other propagule was deposited. Due to their sessile nature in the environment, we provide a few examples that demonstrate which plants have similar strategies but with different final results. One group of compounds that probably best represents this concept are the non-proteinogenic amino acids, used by plants as a response to different kinds of stress [210]. Beyond GABA accumulation under osmotic stress (see above), a plethora of non-proteinogenic amino acids are synthesized by plants. As an example of this, after studying the metabolic response against osmotic stress in tolerant and sensitive varieties, one of the most abundant compounds in roots of the tolerant variety was found to be β-alanine [211]. It acts as a precursor of β-alanine betaine, which acts as osmoprotectant in some halophytic plants [212]. 

Another interesting response is osmoprotection in watermelon; besides the usual accumulation under stress of proline, watermelon complements this by accumulating citrulline. Under stress, the latter represents 21–25% of the total amino-acids in watermelon stem and leaves [213]. These particular responses offer opportunities to use modern omic technologies. Stress-tolerant or sensitive varieties can be compared using metabolomic tools to detect and evaluate the differences arising from the interactive coevolution of plants with their environments [214]. In this way, the search continues for new compounds to further the advance into various fields of biostimulation [215].

### 6.2. New Strategies to Find New Active Biostimulants

In the literature, there are already plenty of approaches to studying differences in tolerance of different crop cultivars, but surprisingly we have not found examples of their use with biostimulants. Here, we discuss some strategies that in our opinion have great screening potential.

Infra-Red thermography is able to detect excess heat emitted from stressed plants [216]. Applying this, Siddiqui et al. [217] demonstrated how the leaf temperature of plants submitted to drought stress increases in a high correlation with relative water content and osmotic potential, and a good correlation with stomatal conductance. Others have used drought-tolerant maize genotypes or salt-tolerant cereals [214,218]. One of the drawbacks of this approach has been the need to use a thermal camera, but recently some papers show how a cheap camera coupled to a smartphone is adequate for infra-red thermography studies [219]. 

The infrared approach can be used within a high-throughput phenotyping platform, such as that described by Kim et al., [218]. This is able to measure plant area, colour, compactness, seed/cereal water content, and photosynthetic efficiency in real time, using image technology and DroughtSpotter. The latter is a lysimeter that assesses water use and transpiration rates based on weight measurements, Phenospex©. Another interesting phenotyping platform measures root and shoot growth ratio in real time [220], which is very useful to take into account in plant breeding against osmotic stress because it normally increases in response to drought [221]. This type of expensive equipment would seem utopic for the majority of researchers but the European Council offers opportunities to use such platforms within the European Plant Phenotyping Network. This leads in interesting directions for new biostimulant development. Rouphael et al. [222] review High-Throughput Plant Phenotyping systems for developing new biostimulants, providing interesting alternatives to perform experiments with transfer from laboratory to field.

## 7. Future Remarks

Plans such as the ‘new green deal’ include a search for field treatments that protect crops without environmental cost. In fact, biostimulants are a promising bet for inclusion in the forthcoming new deal. This has to be faced in a multidisciplinary way, because understanding metabolism under both stress and its alleviation is the key to increased production. In our opinion, this area needs deeper study focused on production and agronomy. In Garcia-Garcia et al. [18], we presented a more detailed review focused on production. After an exhaustive bibliographical search, only 6% of more than 182 papers about biostimulants take into consideration final yield, which is in the end the main purpose of biostimulant utilization. Another important problem with biostimulant research is the exposure to extremely strong stresses, feasible in the laboratory but practically impossible to find in a commercial plantation. A closer model to agricultural reality was our field trial using lettuce as a crop; plants were submitted to stress levels that ensured production but with a considerable impact on it [206]. Although lettuce is a good option for first trials with a biostimulant, it only permits stress protection to be evaluated in terms of vegetative growth. Fruit production is more important in commercial crops, so more research in this latter line is necessary.

## Figures and Tables

**Figure 1 plants-10-00363-f001:**
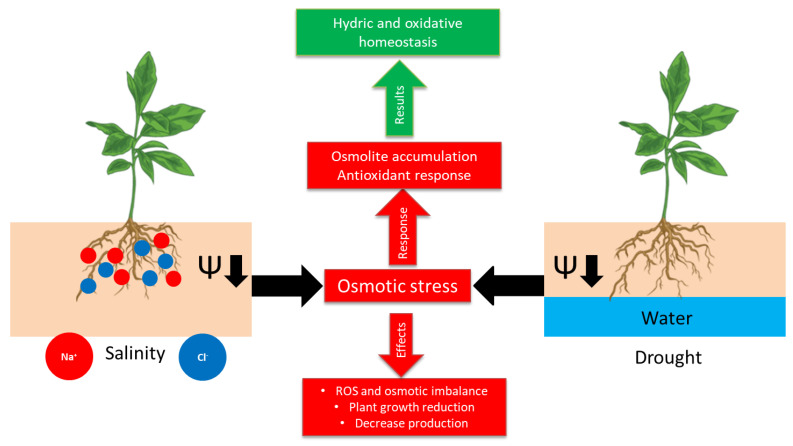
Schematic model of osmotic stress.

**Figure 2 plants-10-00363-f002:**
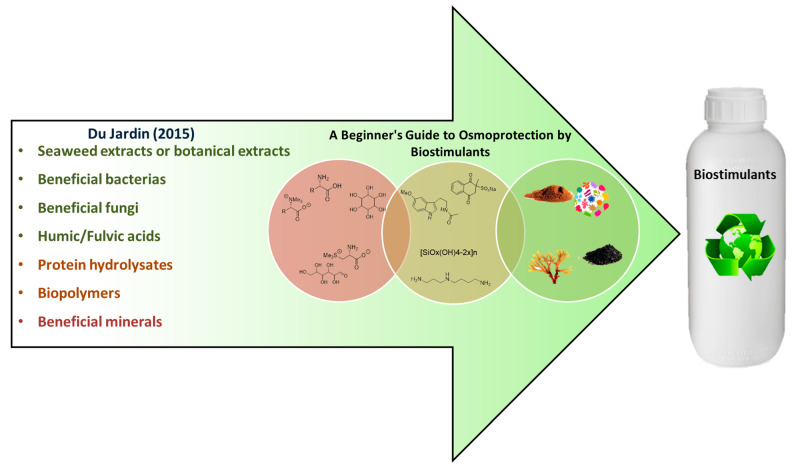
Graphical summary of the components listed in the review: in red, naturally accumulated plant osmoprotectants; in brown, natural and non-natural plant protectant compounds; and in green, hydrolysed biological extracts and microorganisms.

**Figure 3 plants-10-00363-f003:**
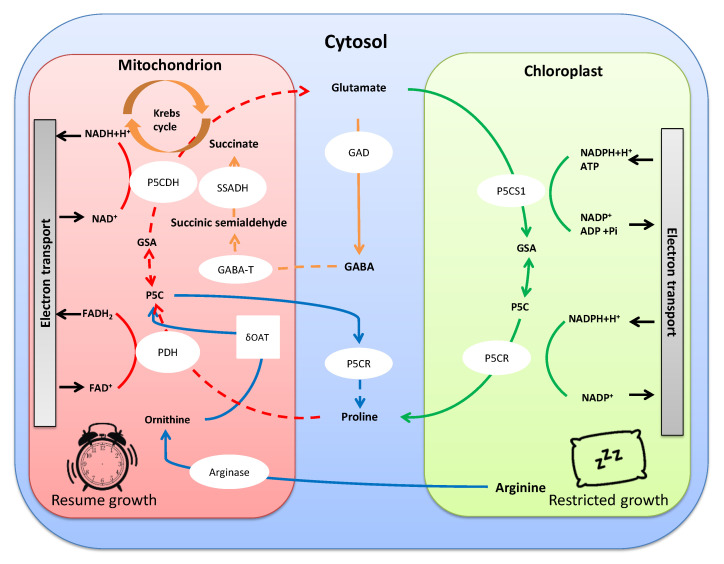
Importance of amino acid anabolism under stress (continuous arrows) to keep the NADPH:NADP^+^ ratio low in the photosynthetic chain, preventing photoinhibition. Amino-acid catabolism (dashed arrows) provides energy for plant growth to resume.

**Figure 4 plants-10-00363-f004:**
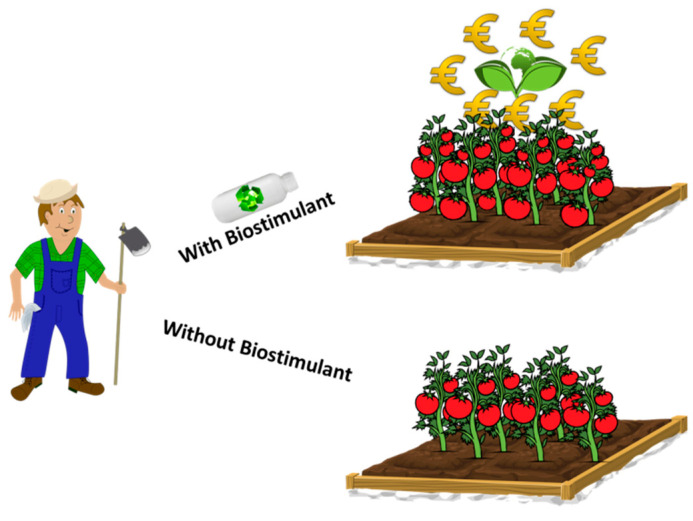
Biostimulants have the potential to increase crop productivity in a sustainable way.

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
