# Peer review of "A Beginner’s Guide to Osmoprotection by Biostimulants"

_plants, 2021, doi:10.3390/plants10020363_

Round 1
Reviewer 1 Report
Manuscript ID plants-1100227 entitled “A beginner's guide to osmoprotection by biostimulants” by Jiménez-Arias et al. reviews some available data on the promising use of biostimulants in current agriculture. In fact, this is a very hot area of research since biostimulant application may help to reduce the effects of abiotic and biotic stresses that cause a huge reduction in crop yields. However, the present MS doesn’t reach its objectives of adding new information on the mechanistic aspect of biostimulants. The review does not also offer a beginner´s guide, since authors present a list of well-known osmoprotectant (proline being the star), but no a clear picture of how these possible biostimulants could be applied in the field.
Moreover, the present review does not seem to add any new relevant information to the several recent reviews published on this hot topic, as Yakhin et al, 2017; Bulgari et al, 2019; Basile et al; Rouphael et al, 2020, and the author´s own recent review García-García et al, 2020)
MS also presents an unclear picture of the classification (Fig 2) of biostimulants that do not add any relevant novelty to the already published classifications. Besides, I do not understand what do other compounds mean in this picture, contrasting to natural compounds (does it means that these other compounds are synthetic (non-natural) ones? or, perhaps, authors mean inorganic, versus organic compounds?. It should be clarified. This undefinition is also applicable to information presented in figures 1 and 3. In figure 1, it appears to indicate that induction of osmoprotectant accumulation does indeed restores plant growth. Authors should be aware that osmolytes accumulation occurs both in resistant and sensitive plant genotypes where it helps survival or tolerance, but there is no direct relationship between accumulation of these substances and restoration of plant growth. On the contrary, some (many) times, the accumulation of osmolytes occurs together with growth arrest. It is not clear to this reviewer how the representation in Fig 3 highlights the role of proline, glutamate, and GABA in osmotic stress tolerance. To do that, authors should point how these metabolites change in response to osmotic stress and how these changes could help to the tolerance to the stress. This is not clear from the presented figure, in which only the well-known biosynthetic pathways of these molecules is shown.
The authors present a concluding remark (conclusion section) that does not reflect the main take-home information presented in the MS, but instead a list of what it should be done in the future. I agree with the authors in that effect of biostimulants in crop yields should be evaluated, with a closer view of actual agricultural practices. However, the data presented here do not fulfill this expectation.
The present text should be better organized and largely improved before its publication in this journal.
Author Response
Please, see attachment

Reviewer 2 Report
The manuscript entitled “A beginner's guide to osmoprotection by biostimulants” describes an interesting review on biostimulants components and their role to increase tolerance to osmotic stress. I found this paper very well composed and rich in references.
However, It still needs a minor revision, following the points listed below, to be fully published.
Line 573-574: which kind of “hard stresses” are you addressing?
Line 574-578: this part of the manuscript seems inappropriate into the conclusion paragraph. You should add in text a small paragraph with this case study.
It has been a pleasure for me to revise this manuscript.
Best regards.
Author Response
Please, see attachment

Reviewer 3 Report
I think the review was prepared very well, in-depth.
Nevertheless, if the Authors wish to add some further notes, I can suggest some additions that may be useful for discussion.
Page 8: Silicon
The thickening of the cuticular tissues due to Silicon application greatly reduces the rate of evapotranspiration with a consequent reduction of possible damage from water imbalances (Ma and Takahashi, 1990). It happened on grapevine, where Silicon application had an important effect on the improvement of berry thickness (greater values), enhancing resistance to fungal diseases too
Page 10. 4.1 Seaweed extracts
Water stress reduces the cytokinin content in xylem tissues and increases the synthesis of abscisic acid. Cytokinins, in fact, are key phytohormones that not only regulate plant growth and development but also mediate plant tolerance to drought stress (Hai et al., 2020).
The use of algae, in addition to acting positively on the synthesis of chlorophyll, keeps the concentration of cytokinins high and hinders the synthesis of ABA.
Treatments with Ascophyllum nodosum extracts provide drought stress tolerance to treated plants by reducing stomatal conductance and cellular electrolyte leakage in water-stressed plants over time, and maintaining a high leaf turgor and stem angle (Norrie, 2016). This can cause a direct improvement of the physiological status of plants, as observed on a grapevine trial, where foliar application of a product based on algal extracts (from red seaweeds, rich in betaine and Iodine) enhanced both leaf greenness and chlorophyll content, as well as the vigor of plants in terms of biomass photosynthetically active
Hai N.N., Chuong N.N., Tu N.H.C., Kisiala A., Hoang X.L.T., Thao N.P. (2020) - Role and Regulation of Cytokinins in Plant Response to Drought Stress. Plants 2020, 9, 422. https://doi.org/10.3390/plants9040422
Ma, J., Takahashi, E. (1999) - Effect of silicon on the growth and phosphorus uptake of rice. Plant Soil 126, 115–119. https://doi.org/10.1007/BF00041376
Norrie J. (2016) - Ascophyllum nodosum extracts: gifts from Poseidon to Theoi Georgikoi (the Greek gods of agriculture). Acta Hortic. 1148. ISHS 2016. DOI 10.17660/ActaHortic.2016.1148.1
Author Response
Please, see attachment

Round 2
Reviewer 1 Report
The present version of the revised MS has consistently corrected the several flaws of the previous version, thus it is now suitable for its publication.